# COVID-19 Diagnostics Outside and Inside the National Health Service: A Single Institutional Experience

**DOI:** 10.3390/diagnostics11112044

**Published:** 2021-11-04

**Authors:** Lukasz Fulawka, Aleksandra Kuzan

**Affiliations:** 1Molecular Pathology Centre Cellgen, 50-353 Wroclaw, Poland; biuro@cellgen.pl; 2Department of Biochemistry and Immunochemistry, Wroclaw Medical University, 50-367 Wroclaw, Poland

**Keywords:** COVID-19, SARS-CoV-2, diagnostics, prevalence

## Abstract

The COVID-19 epidemic has been going on continuously for more than 1.5 years. Fast and reliable diagnosis is a key component of an outbreak response strategy. Our goal is to present the statistics from one of the diagnostic points of a large city in Poland. Swabs of the throat or nasopharynx of people reporting for molecular diagnostics of SARS-CoV-2 presence were taken. CE-IVD-certified RNA isolation and RT-PCR assays were used. According to our data, the prevalence of SARS-CoV-2 infection in the examined population equaled 14.7%; however, large differences were observed depending on where the sampling point was located: as much as 50.3% of positive results for samples collected at a stationary point, 36.2% for samples from inpatients and hospital staff, and only 8.9% for samples from patients whose test was paid by their employer. The age structure of the infected population was fairly even, with a slightly higher number of people over 50 years of age. Men were examined more often, but it was among women that a higher percentage of infection was recorded. Every fifth test was performed for a foreigner, but compared to Poles, a much lower incidence of infection was found in these samples. We conclude that due to the high prevalence of infection in patients from social care centers and in those referred to hospitals, it is recommended that a special sanitary regime is followed in those settings. We will evaluate the effectiveness of vaccinations, expecting that the coming months bring positive changes in the statistics on prevalence.

## 1. Introduction

At the beginning of January 2020, the first scientific papers describing the coronavirus disease 2019 (COVID-19) caused by severe acute respiratory syndrome coronavirus 2 (SARS-CoV-2) were published [1]. More than a year and a half after that, we already have a large amount of information about the pathomechanism of COVID-19, its epidemiology, health consequences, and even prevention options related to the developed vaccines.

Our goal is to present the epidemic from a different perspective, to show a lesson from the practice of a private diagnostic laboratory, which from May 2020 to the present time, has conducted diagnostics in a large city in Poland. Our aim was to present the dynamics of the prevalence of COVID-19 during the epidemic in a large city in Central Europe as well as the profile of customers who used COVID-19 diagnostics within and outside the National Health Fund.

SARS-CoV-2 diagnostics is based on a variety of tests; however, the WHO has issued a statement that suspect cases should be screened for the virus with nucleic acid amplification tests (NAAT), such as real-time reverse-transcription polymerase chain reaction (RT-PCR), reverse-transcription loop-mediated isothermal amplification (RT-LAMP), magnetic nanoparticle (MNPs)-based methods, and others, and recommend the use of the RdRp, E, N, and S genes in different combinations. Other methods, such as a serological analysis, can only be used as an aid, but do not confirm infection. RT-PCR remains a reference method with advantages such as high sensitivity, specificity, and compatibility with automation and multipanels [2]. In this paper, we present our results from RT-PCR only, although the company also performed cassette antigen tests.

## 2. Patients and Methods

### 2.1. Study Group

Samples of nasopharyngeal swabs donated in order to perform a diagnostic test for the presence of SARS-CoV-2 RNA were analyzed. The study was performed at the private diagnostic company Molecular Pathology Center Cellgen. All participants signed an informed consent to participate in the research study. With personal data, patients provided information only about their sex, age, and place of residence.

Some of the patients, for whom it was possible, were also categorized in terms of where and how they performed the examination. We distinguished:(1)patients who came to the drive-thru point (they paid for the test themselves)—10,676 patients,(2)patients from private companies where the employer paid for the test—6873 patients,(3)outpatients and clinic staff, for whom the payer was the National Health Fund or the founder—484 patients,(4)patients coming to a stationary collection point, for whom the payer was the National Health Fund—312 patients,(5)patients referred to a sanatorium, for whom the payer was the National Health Fund—30 patients,(6)social care centers, where the payer was the National Health Fund—34 patients,(7)inpatients and hospital staff—3939 patients.

This study analyzed samples collected between 1 May 2020 and 31 August 2021.

### 2.2. RNA Isolation

RNA isolation was performed manually or automatically. Manually, between 1 May 2020 and 20 September 2021, with CE-IVD-certified kits: (1) EliGene^®^ Viral RNA/DNA FAST Isolation Kit (Elisabeth Pharmacon Ltd., Brno-Zidenice, Czech Republic), (2) Invisorb^®^ Spin Universal Kit (Invitek Molecular GmbH., Berlin, Germany), or (3) PowerPrep™ Viral DNA/RNA Extraction Kit (KogeneBiotech Co., Ltd., Seoul, Korea), following the manufacturer’s recommendations. Automatically, between 21 September 2020 and 31 August 2021 with CE-IVD-certified kits and workstations: (1) TANBead^®^ Nucleic Acid Extraction Kit (Taiwan Advanced Nanotech Inc., Taoyuan City, Tajwan) using TANBead^®^ Nucleic Acid Extraction System Maelstrom 4800 (Taiwan Advanced Nanotech Inc., Taoyuan City, Taiwan) and (2) ANDiS Viral RNA Auto Extraction & Purification Kit (3D Biomedicine Science & Technology Co., Ltd., Shanghai, China) using Automated Nucleic Acids Extraction System ANDiS 350 (3D Biomedicine Science & Technology Co., Ltd., Shanghai, China), following the manufacturer’s recommendations. Real–Time PCR CE-IV-certified Vitassay qPCR SARS-CoV-2 kit (Vitassay Healthcare, Huesca, Spain) and Viasure SARS-CoV-2 Real-Time PCR Detection Kit (Certest Biotec S.L., Zaragoza, Spain) was used for the qualitative detection of SARS-CoV-2 RNA by real-time amplification of specific conserved fragments within ORF1ab and N viral genes. The procedure was conducted on a BioRad CFX96 instrument with the thermal cycler BioRad C1000 (Bio-Rad Laboratories, Inc., Hercules, CA, USA), according to the manufacturers’ recommendation. Positive and negative samples were included at each run.

### 2.3. Statistical Analysis

The results are presented in the form of descriptive statistics with the use of Microsoft Excel software. The statistical analysis was conducted with Statistica software (version 13.3, StatSoft; TIBCO Software Inc., Palo Alto, CA, USA). Chi square and ANOVA tests were used.

## 3. Results

Overall, the number of tests performed between 1 May 2020 and 31 August 2021 was 31,964, of which 3687 were positive, 27,883 were negative, 360 were inconclusive, and 14 were non-diagnostic. A graphical representation of this result with the calculated percentage of the total study is shown in Figure 1A.

Some patients were examined more than one time. When analyzing the prevalence, understood as whether a given patient was infected with SARS-CoV-2, excluding inconclusive and non-diagnostic results, we obtained different results from the above-mentioned ones. In fact, in that case, the percentage of positive results was 14.7% (*n* = 3592), and the percentage of negative results was 85.3% (*n* = 20,853). Non-diagnostic and inconclusive results were rejected for this summary (see Figure 1B).

The data on the number of confirmed infections per unit of time are presented in Figure 2. It can be seen that the majority of infections were in October 2020, and a large number of infections were also found in March 2021 and November 2020. The cumulative data are presented in Figure 3. It can be seen that the percentage of positive tests increased in proportion to the number of tests performed, and the jumps on the line showing the positive cases coincide with a sharp increase in the number of tests performed.

The age structure of the examined participants was also analyzed. The mean and median age were 40 and 38, respectively (*n* = 19,262 cases with known age, calculated using PESELs—Polish ID numbers). There was a slight difference between positive (mean 42, median 39, *n* = 3361) and negative cases (mean 40, median 38, *n* = 15,901), which was statistically significant (one-way ANOVA, *p* < 0.001).

Graphically, the data are presented in Figure 4. However, there was no drastic difference in the prevalence depending on the age group—over 12% of prevalence was observed in each of them. In the analyzed group, the prevalence was slightly higher in the group of women. Interestingly, more men had the test ordered—13,419 men versus 9747 women, of which 1920 and 1616 had a positive result, respectively. The difference in prevalence between men (14.3%) and women (16.6%) was statistically significant (Pearson, Chi-square: *p* < 0.01). The comparison regarding sex is presented in Figure 5A.

Among the examined persons, 19,262 were Polish citizens (with valid PESEL numbers), while 5183 were foreigners. There was a noticeable difference in the prevalence between the two groups—among Poles it was 17.4% (*n* = 3361), while among foreigners only 4.5% (*n* = 231) (chi-square: *p* < 0.05). These data are graphically presented in Figure 5B.

By comparing the data on how the test was performed and who the payer was, we found that:(1)patients who came to drive thru points: 10,610 people, including 11.5% (*n* = 9390) positive cases(2)patients from private companies, where the employer paid for the examination: 6088, including 8.9% (*n* = 544) positive cases(3)outpatients and clinic staff: 555 patients, including 16% (*n* = 89) positive cases(4)stationary collection point where the payer was the National Health Fund: 312 patients, including as much as 50.3% (*n* = 157) positive results(5)patients before admission to a sanatorium, where the payer was the National Health Fund: 30 patients, of which only 1 had a positive result (3.3%),(6)people from social care centers where the payer was the National Health Fund: 34 patients, including 20.6% (*n* = 7) positive cases,(7)inpatients and hospital staff for whom the payer was the National Health Fund or the founder: 3942 patients, of which 36.1% (*n* = 1424) were positive.

The above differences were statically significant (Chi-square: *p* < 0.05).

To sum up, 4873 tests were ordered by the National Health Fund, 6088 tests were paid by the employer, and the largest number of tests—10,610—were paid by patients. For the rest of the tests, it was impossible to determine the payer. It should be noted that there were many more laboratories in the city that performed tests ordered by the National Health Fund.

The results of the prevalence of SARS-CoV-2 infection in the above-mentioned contexts are graphically presented in Figure 6.

## 4. Discussion

As of 11 October 2021, there were 2,923,304 cases of CODIV-19 infection in Poland, including 75,869 fatal cases [3]. At the beginning of the interpretation of the results, it should be noted that a large proportion of people infected with SARS-CoV-2 are not included in any statistics. This is partly the case for asymptomatic patients who have no motivation to take the test. This also partially applies to people with mild or moderate symptoms of infection who do not have access to the test or refrain from doing so for other reasons. To date, surveillance data following viral RNA detection have not been able to determine the extent of asymptomatic infection, with estimates ranging from very low (6%) to very high (96%), depending on the population studied and the period assessed [4].

The age structure of people infected with SARS-CoV-2 in this study is quite typical and does not differ from those reported in other populations [5,6]. The relationship between the prevalence and sex is also the same as in other similar studies—women are infected more often [7,8]. Perhaps, that result is observed because a large proportion of the tested subjects were female nurses. Perhaps, the susceptibility to SARS-CoV-2 infection is the same for both sexes, and the observed effect is, therefore, an artifact (Raciborski et al. 2020). However, this hypothesis should be verified, which was impossible in the case of this project. Moreover, it should be borne in mind that although the rate of infection by COVID-19 seems to be higher in women, the probability of severe course and death is much higher for men [7,8]. It is suspected that the causes of this phenomenon are related to differences in sex hormones, immune effectors, and sex-specific differences in behavior [9].

Some groups of patients in our study had a relatively low prevalence, such as patients using the drive-thru test points. Our observations suggest that these were rather asymptomatic patients, who came for testing, for example, to obtain a certificate for an airline. Drive-through testing is a logistically quite simple, safe, and efficient method to collect specimens, preferred by many members of the public, as it minimizes the risk of infection as a result of attendance at a medical facility [10].

It is noteworthy that the prevalence was very high—over 50%—among patients reporting to the stationary collection point. In this group, patients were rather symptomatic, as the presence of symptoms was generally required to be referred by a physician for a molecular test. In any case, the high incidence rate exceeded the expected rates for influenza or other similar viral or bacterial diseases. In Poland, in the years 2020/21, we have observed an extraordinary decrease in the incidence of influenza, similar to other countries [11].

It is worth noting that molecular tests detecting the presence of SARS-CoV-2 cost about PLN 500, which is about 10% of the average monthly salary in Poland [12]. Nevertheless, in the Cellgen laboratory, nearly half of the tests were actually paid for by the patient. Unfortunately, these results do not determine whether the patients preferred to pay themselves because they did not have access to the service paid by the National Health Fund, or whether they had access to the paid service but wanted to speed up the receipt of the result. In a certain percentage, which is impossible to estimate, these were tests for travel purposes that required the payment of the costs of the test.

A limitation of our study is that we did not separate the participants from clinics and hospitals into patients and staff. We can see, however, that every fifth tested person at the clinic and every third tested person in the hospital was infected with the SARS-CoV-2 virus; this seems to be a very large percentage, indicating that efforts to obtain sterility in these locations need still to be intensified.

We also observed a very high percentage of cases in social care centers. In our study, this value was 20.3%, whereas, in another similar study, conducted only during the first two months of the epidemic in Poland, it was 13.3% [8]. This was expected, in connection with the difficulties that are often encountered by institutions responsible for the long-term care for people, especially the elderly. There is a shortage of personal protective equipment, difficulties, or even the inability to enforce social distancing and a shortage of staff [13]. We conclude that this is a context particularly prone to the devastating effects of a pandemic, therefore deserving special care.

A relatively large number of tests were paid by the employers—nearly 7000 out of a pool of 31,570—which proves that some employers were interested in the health of their employees and in ensuring the continuity of work in their companies in order to counteract economic losses observed around the world [14]. As many as 75% of COVID-19 patients surveyed in Poland in home isolation became infected at work [15]. This shows how important responsible management by the employer is and that a company’s policy has a large impact on the prevalence of infection in its employees.

The city of Wrocław, where the project was carried out, is one of the largest cities in Poland—it has nearly 650,000 inhabitants. From the point of view of the interpretation of this data, it is also important that this city has an airport as well as 10 public and many private universities, many of which provide education for foreigners. It also has a very extensive economic infrastructure that allows the employment of foreigners. Nevertheless, the result of such a large share of tests for foreigners is quite surprising. The low percentage of positive test results for foreigners can probably be explained by the fact that these participants probably started testing mainly before traveling to another country in order to obtain a certificate. A large proportion of foreigners were also routinely examined by their employers.

Poland introduced extensive anti-epidemic measures relatively early, in order to slow down the spread of the disease. Already on 2 March 2020, i.e., earlier than the first case of COVID-19 infection in Poland, a Crisis act was introduced, that is, a Law on special arrangements for the prevention of and fight against COVID-19 [16,17]. In Poland, there were three waves of the pandemic—the first one in spring 2020, the second one in autumn 2020, the third one in spring 2021. As it can be seen, the first wave was imperceptible in statistical data (see Figure 3 and Figure 4), while the second and third waves were quite clear. However, strict restrictions during the so-called “first wave” (obligation to wear masks, closure of schools, universities, shops, etc.) seemed to result in the fact that there were not many positive cases in spring 2020.

Prevention in Poland, apart from the restrictions regulated by law, has also consisted in introducing vaccinations as early as possible. The first vaccination day in this country was 27 December 2020 [18]. By 11 October, 52.61% of Poles were vaccinated, including 51.7% fully vaccinated [19]. Unfortunately, the vaccination effect is not yet visible in the COVID-19 incidence statistics (see Figure 4). The reduction in summer infections is seasonal, similar to what was seen in the summer of 2020, when there was no vaccine yet. The next few months will reveal how successful the vaccines have been. We are afraid that vaccination of nearly half of the population will be insufficient and that we will observe an increase in infections, i.e., a “fourth wave”. The Polish government is doing a lot to increase the percentage of vaccinated people: it has organized, among others, information campaigns and contests with prizes, has introduced advertisements on TV and other mass media, has launched touring vaccination points and vaccination in pharmacies and shopping malls [18,20].

## 5. Summary and Conclusions

As the pandemic is still ongoing, there is a compelling need to accumulate statistical data and evaluate the full extent of the consequences of the COVID-19 pandemic, including its social, economic, and political impacts, in Poland. Such statistics help to monitor the effectiveness of government restrictions and vaccination. Certainly, the project would be much more informative if, apart from basic data, the respondents provided data on whether they had symptoms and comorbidities, were vaccinated—if so, with which vaccine—traveled recently, etc. However, the observations showed that that patients are tired of the COVID-19 questionnaires they have to complete in various circumstances and are reluctant to consent to participate in a study if they have to answer a time-consuming questionnaire.

This analysis is unique, to our knowledge. No study has presented, for example, statistics showing what proportion of infected people paid themselves for a test, how many of them used stationary diagnostic points or drive-thru points, and how many of them were foreigners. Although only basic statistics are presented here, they bring a lot of knowledge about diagnostics in Poland and the patients who use it. We can conclude that people living in this city are willing to incur the high costs of molecular tests, that the number of positive results is lower for patients tested in clinics than for those who are tested in hospitals, that foreigners in this city are relatively often tested but they have positive test results less often than Poles. We plan to continue anaylzing samples in the near future, expecting the epidemic to be extinguished.

## Figures and Tables

**Figure 1 diagnostics-11-02044-f001:**
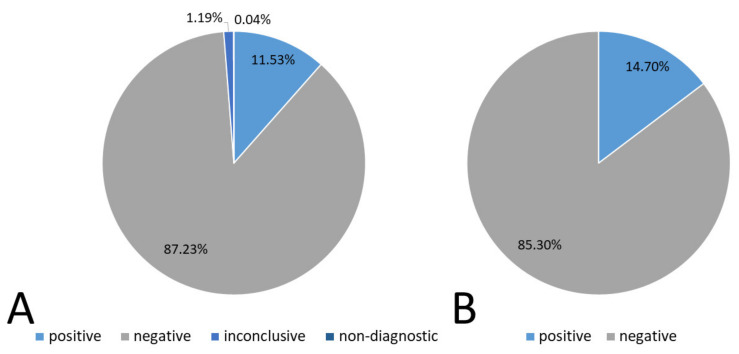
Total tests results including inconclusive and non-diagnostic results (**A**). Prevalence of SARS-CoV-2 infection in the examined population (**B**).

**Figure 2 diagnostics-11-02044-f002:**
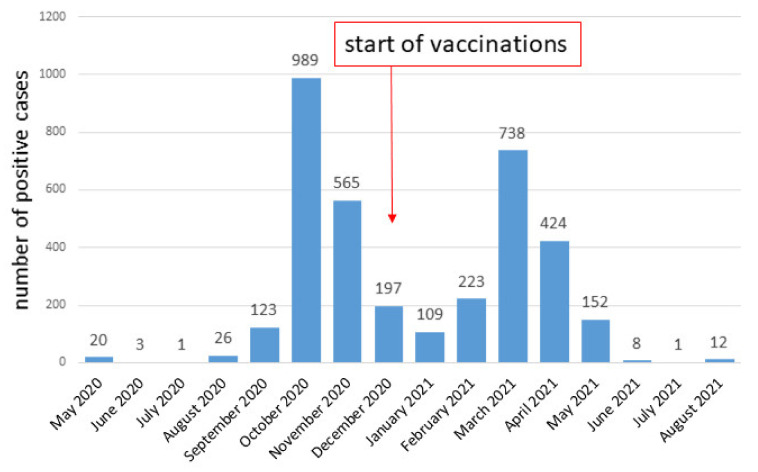
Monthly SARS-CoV-2 infection cases in the examined population.

**Figure 3 diagnostics-11-02044-f003:**
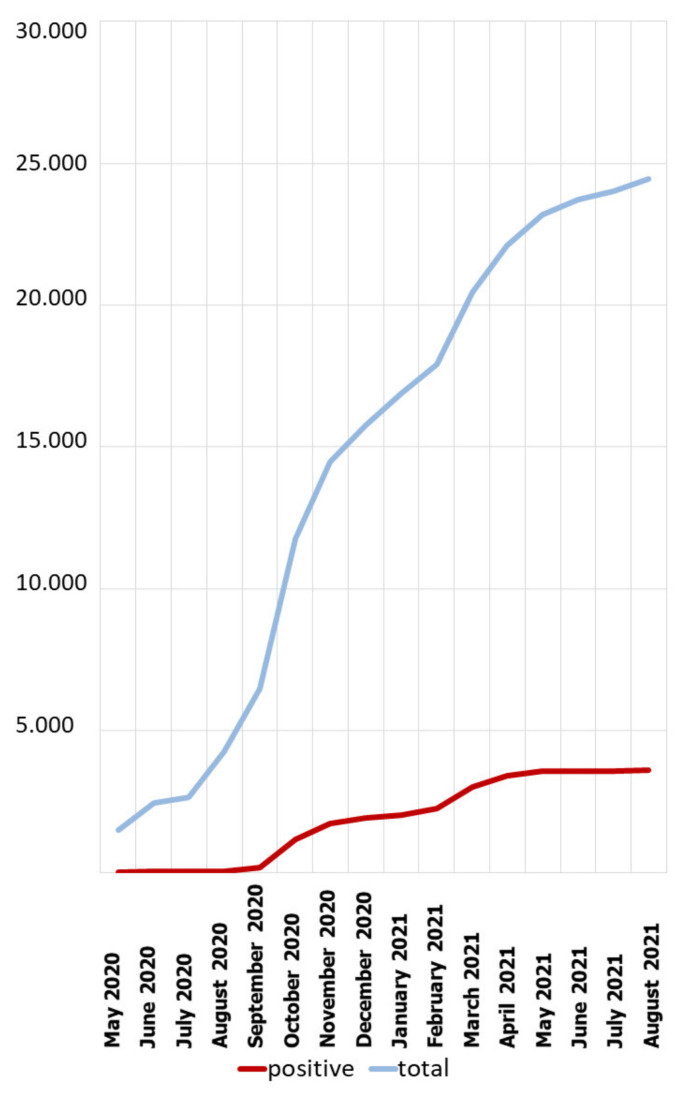
Total cumulative number of tests conducted over time.

**Figure 4 diagnostics-11-02044-f004:**
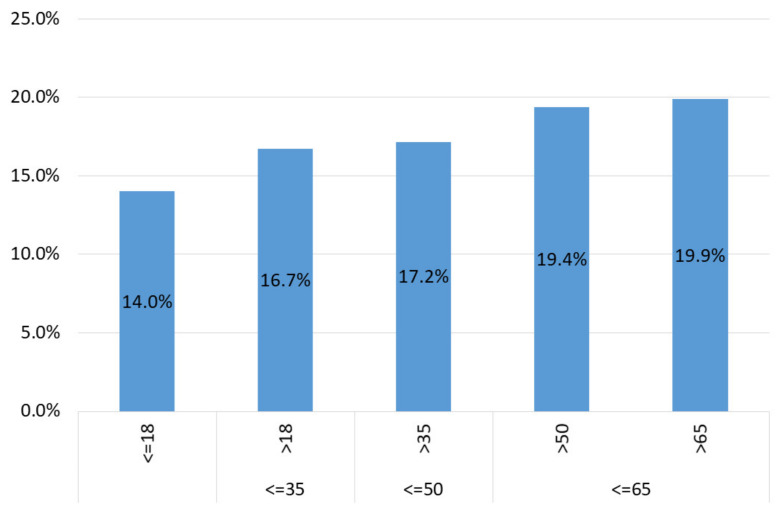
Prevalence of SARS-CoV-2 infection depending on age.

**Figure 5 diagnostics-11-02044-f005:**
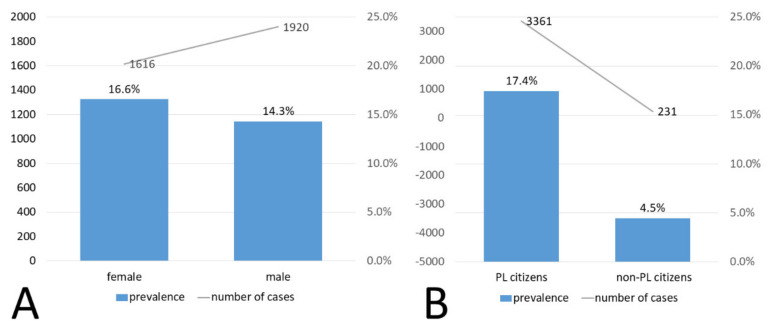
Prevalence of SARS-CoV-2 infection depending on sex (**A**); Prevalence of SARS-CoV-2 infection depending on citizenship (**B**).

**Figure 6 diagnostics-11-02044-f006:**
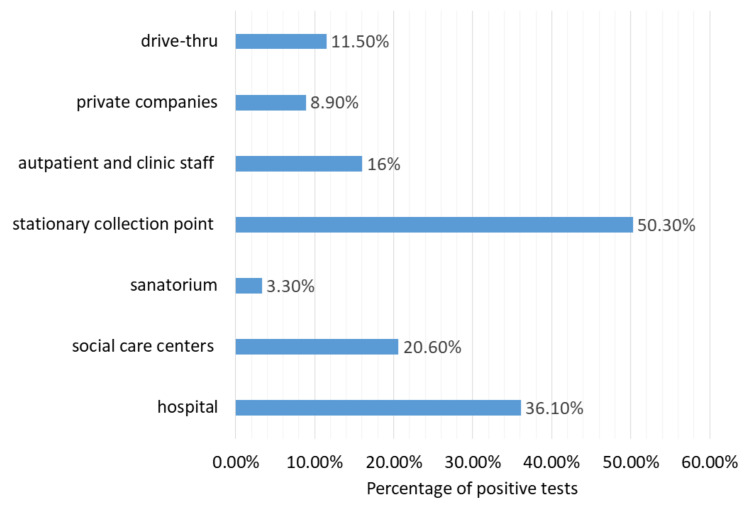
Prevalence of SARS-CoV-2 infection depending on the source of the sample.

## Data Availability

Data supporting the reported results can be obtained from the corresponding author upon request.

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
