# Peer review of "COVID-19 Diagnostics Outside and Inside the National Health Service: A Single Institutional Experience"

_diagnostics, 2021, doi:10.3390/diagnostics11112044_

Round 1

Reviewer 1 Report

This paper describes the results of Covid testing in a larger city in Poland. The results are straightforward and reasonably well presented. They are in line with those of similar reports from other countries. They are neither unexpected nor very exciting. Still I recommend publication of the paper since it is of interest to know how the pandemic develops in different countries.

I have a few questions:

I do not understand the difference between panels A and B. The difference in prevalence in the panels seems greater than accounted for by non-diagnostic and inclusive results.

What does non-diagnostic mean?

It is claimed that the difference in gender prevalence could be due to a large proportion of nurses. What was the proportion of nurses in the total cohort?

Given the fact that the results largely confirm published observations, I recommend that the Discussion be shortened

Reviewer 2 Report

The authors' goal was to present the dynamics of the prevalence of COVID-19 during the epidemic in a city in Poland.  They concluded that people are willing to incur high costs of molecular tests, that patients tested at clinics have less positive results than those who are tested at hospitals, among other conclusions.

I consider approving the manuscript, after correcting a few grammatical errors. In addition, the figures should be replaced by higher resolution.
